# Voxel Proposal Network via Multi-Frame Knowledge Distillation for Semantic Scene Completion

Lubo Wang[1] *    Di Lin[1] *    Kairui Yang[1]    Ruonan Liu[2] †    Qing Guo[3]    Wuyuan Xie[4]

Miaohui Wang[4]    Lingyu Liang[5][6]    Yi Wang[4]    Ping Li[7]

[1]College of Intelligence and Computing, Tianjin University    [2]Shanghai Jiao Tong University
[3]IHPC and CFAR, Agency for Science, Technology and Research, Singapore
[4]Shenzhen University    [5]Pazhou Lab    [6]South China University of Technology
[7]The Hong Kong Polytechnic University
wanglubo@tju.edu.cn   di.lin@tju.edu.cn

## Abstract

Semantic scene completion is a difficult task that involves completing the geometry and semantics of a scene from point clouds in a large-scale environment. Many current methods use 3D/2D convolutions or attention mechanisms, but these have limitations in directly constructing geometry and accurately propagating features from related voxels, the completion likely fails while propagating features in a single pass without considering multiple potential pathways. And they are generally only suitable for static scenes and struggle to handle dynamic aspects. This paper introduces Voxel Proposal Network (VPNet) that completes scenes from 3D and Bird's-Eye-View (BEV) perspectives. It includes Confident Voxel Proposal based on voxel-wise coordinates to propose confident voxels with high reliability for completion. This method reconstructs the scene geometry and implicitly models the uncertainty of voxel-wise semantic labels by presenting multiple possibilities for voxels. VPNet employs Multi-Frame Knowledge Distillation based on the point clouds of multiple adjacent frames to accurately predict the voxel-wise labels by condensing various possibilities of voxel relationships. VPNet has shown superior performance and achieved state-of-the-art results on the SemanticKITTI and SemanticPOSS datasets.

## 1 Introduction

Understanding 3D scenes based on LiDAR point clouds is essential for tasks like autonomous driving. However, due to the limitations of LiDAR sensors and the occlusion of instances by themselves or other instances in the real world, including large-scale information in the point clouds poses a significant challenge to understanding 3D scenes.

Semantic Scene Completion (SSC) aims to simultaneously infer a scene's occupancy and semantic information based on point clouds using deep learning. Several methods [1; 2; 3; 4; 5; 6; 7; 8; 9; 10; 11; 12; 13; 14; 15; 16; 17; 18] use convolutions to complete the partial scene. Some completion methods [19; 20; 21; 22] heavily rely on diverse attention mechanisms as the attention mechanism can capture spatial relationships and update the features. The diffusion model [23] also applies to the

---

*Co-first authors.
†Corresponding author.

completion task. These methods have significantly improved the performance of static single-frame-based semantic scene completion. However, they still suffer from extreme geometric incompletion due to the large-scale information loss of point clouds. Moreover, these methods ignore the regional distraction and voxel semantic uncertainty that arises from the information loss and the complex relative motion of instances in dynamic point cloud sequences.

Our paper introduces a new method for completing from both Bird's Eye View (BEV) and 3D perspectives. We propose confident voxels that show possibilities for voxels and implicitly capture the uncertainty of voxel-wise labels. Our method, Voxel Proposal Network (VPNet), includes the Confident Voxel Proposal (CVP) and Multi-Frame Knowledge Distillation (MFKD). We present the overview architecture of VPNet in Figure 1. The BEV branch completes from the BEV perspective using 2D convolutions to ensure global reasonableness and comprehensiveness of completion. The 3D branch consists of segmentation and completion subnetworks, which perform completion under the guidance of rich semantic contexts and optimize local details and accuracy of completion.

In the 3D branch, CVP learns multiple arrays of offsets for occupied voxel coordinates and features to compute confident voxel coordinates and perform long-range feature propagation like [24] within its branches. Then, CVP uses the confident voxel coordinates to propose confident voxels and constructs confident feature maps, which suggest various possibilities of voxel semantic labels. Finally, we integrate the confident feature maps as augmented feature maps for completion using multi-branch fusion, which condenses the proposed possibilities from the inner-frame branches.

VPNet has a multi-frame network that generates enhanced feature maps for multiple frames using CVPs. It condenses these feature maps into the branches of the CVP in the single-frame network, enabling each branch to create a similar semantic to the corresponding point cloud frame. VPNet condenses the combined enhanced feature map of multiple frames into the single-frame network, further improving the semantics. This process condenses the various possibilities in multi-frame to single-frame networks and affords the opportunity to learn to infer the lost details of each frame in contrast to other KD methods [25; 26; 27; 16].

We evaluate the effectiveness of VPNet on the SemanticKITTI [28] and SemanticPOSS [29] datasets, where we achieve state-of-the-art performances on the semantic scene completion task.

## 2 Related Work

### 2.1 Semantic Scene Completion

The current approaches for SSC rely on convolution, attention, or diffusion models. For example, SSCNet [1] utilizes dilated convolution to enhance the feature map, while LMSCNet [2] applies 2D U-Net and 3D segmentation heads for multi-resolution completion. ESSCNet [3] employs spatial group convolution and sparse convolution to group voxels, and UDNet [4] incorporates UD block in 3D U-Net to efficiently fuse encoder and decoder features. Furthermore, SSA-SC [5] uses a semantic segmentation network to assist completion from BEV, JS3CNet [6] employs dense 3D and graph convolution to link point cloud and voxel features, and Symphonies [19] adopts deformable cross-attention to generate voxel features from the multi-scale depth and RGB images. Lastly, VPDD [23] utilizes the diffusion model to remove noise and complete the scene.

These methods have geometry construction and feature propagation limitations as they assume a static perspective and disregard the semantic uncertainty in dynamic sequences. To address this, we propose a method that leverages confident voxels to model the semantic uncertainty and employs multi-frame distillation to enhance the uncertainty modeling.

### 2.2 Knowledge Distillation

Knowledge distillation transfers knowledge from the teacher to the student network. Smaller3D [25] distills at different levels. PointDistiller [30] proposes local distillation. PVD [26] adopts super voxel partition to utilize geometric information better. S2M2-SSD [31] distills multi-modal knowledge to network with point cloud as input. CMKD [32] distills from the network with the point cloud as input to the network with the monocular image as input. 2DPASS [33] proposes multi- to single-modal distillation that fuses point cloud and image features and distills fused features to the student network. SMF-SSD [34] distills multi-frame knowledge to a single-frame network at three levels. M2SKD [35]

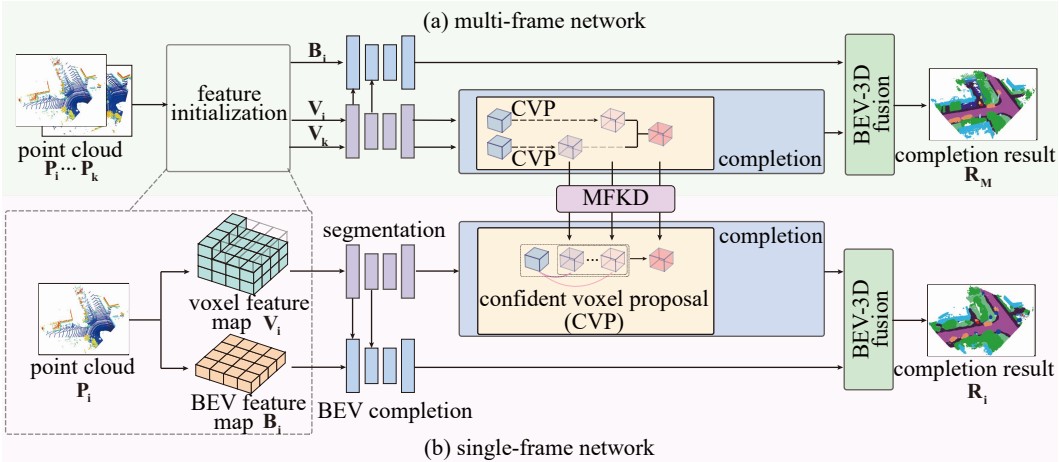

Figure 1: The architecture of VPNet. It consists of BEV and 3D completion branches. CVP in the 3D branch proposes confident voxels to present possibilities for voxels and model the semantic uncertainty of voxels implicitly. Moreover, we construct a multi-frame network and employ MFKD to enhance the accuracy of uncertainty modeling. We represent free voxels as transparent.

performs distillation for difficult categories from multi- to single-frame. 3D-to-BEV [27] achieves distillation from 3D to BEV view. SDSeg3D [36] distills from data-augmented teacher to student without augmentation to improve robustness. In contrast, we adopt multi-to-single frame distillation to extract accurate semantic information.

## 2.3 Point Cloud Sequence Learning

The information within adjacent point cloud frames is complementary. This understanding forms the basis of our research. M2SKD [35] and SMF-SSD [34] fuse aligned multi-frame point cloud for distillation, specially M2SKD [35] only fuses complex samples. TemporalLatticeNet [37] adopts LSTM and GRU to capture temporal relationships better. MarS3D [38] builds Motion-Aware Feature Learning to extract motion instance features. TemporalLidarSeg [39] and MemorySeg [40] employ a Memory mechanism to fuse the features with other frames. Meta-RangeSeg [41] uses Meta-Kernel [42] to aggregate spatial-temporal features. SpSequenceNet [43] proposes Cross-frame Global Attention and Local Interpolation to fuse features. P4Transformer [44] designs Point 4D Convolution to capture the spatial-temporal relationship. PST-Transformer [45] extract spatial-temporal features in a decoupled-joint manner. Moreover, SVQNet [46] splits historical points into voxel-adjacent neighborhoods and historical contexts to complete local and global information. While commendable, existing methods often struggle to fuse point cloud information efficiently. They cannot assign different weights to point clouds, highlighting the need for a more comprehensive solution. We construct a multi-frame network by fusing the feature maps with weighted fusion to guide the single-frame network by distillation.

## 3 Method Overview

We present VPNet and pipeline of CVP and MFKD in Figure 1 and 2. In single-frame network, given point cloud $\mathbf{P}_i \in \mathbb{R}^{N_i \times 4}$, we process it with shared MLP and get 3D voxel feature map $\mathbf{V}_i \in \mathbb{R}^{L \times W \times H \times C}$ and BEV feature map $\mathbf{B}_i \in \mathbb{R}^{L \times W \times C}$ (see Figure 1(b)), $N_i$ is the number of points, $C$ is the channel number of $\mathbf{V}_i$ and $\mathbf{B}_i$. We pass $\mathbf{B}_i$ through BEV completion branch and get completed BEV feature map $\mathbf{F}_i^{bev} \in \mathbb{R}^{L \times W \times H \times C'}$, $C'$ is the channel number of $\mathbf{F}_i^{bev}$. Then, we pass $\mathbf{V}_i$ through segmentation and completion subnetwork in the 3D completion branch. Given semantic embedded feature map $\mathbf{S}_i \in \mathbb{R}^{L \times W \times H \times C'}$ produced by segmentation subnetwork, **Confident Voxel Proposal** (CVP) learns $Q$ groups of offsets $\{\mathbf{D}_i^q \in \mathbb{R}^{J_i \times 3} \mid q = 0, 1, \ldots, Q-1\}$ for occupied voxel coordinates $\mathbf{M}_i \in \mathbb{R}^{J_i \times 3}$ and features $\mathbf{U}_i \in \mathbb{R}^{J_i \times C'}$ in $\mathbf{S}_i$ to compute confident voxel coordinates and features, it builds confident feature maps $\{\mathbf{E}_i^q \mid q = 0, 1, \ldots, Q-1\}$ with confident voxels and $\mathbf{S}_i$ with $Q$ branches in CVP. And CVP produces an augmented feature map $\mathbf{A}_i \in \mathbb{R}^{L \times W \times H \times C'}$ by

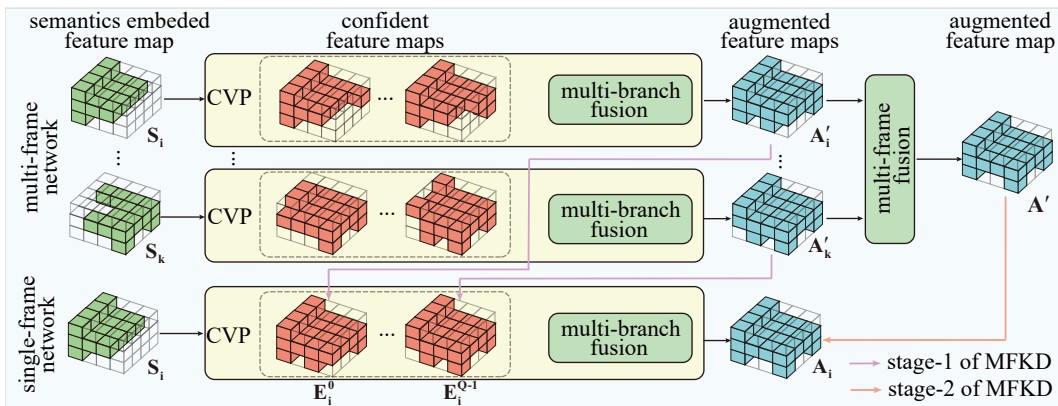

Figure 2: The pipeline of CVP and MFKD. The semantics feature maps are produced with a segmentation subnetwork in the 3D branch.

fusion of $\{\mathbf{E}_i^q \mid q = 0, 1, \ldots, Q-1\}$, $J_i$ is the number of occupied voxels. After that, we pass $\mathbf{A}_i$ through other parts of completion subnetwork as completed 3D feature map $\mathbf{F}_i^{3d} \in \mathbb{R}^{L \times W \times H \times C'}$ (see Figure 1(b)). Finally we fuse $\mathbf{F}_i^{bev}$ and $\mathbf{F}_i^{3d}$ with BEV-3D Fusion to maintain final completion result $\mathbf{R}_i \in \mathbb{R}^{L \times W \times H \times C''}$ where $C''$ indicates the number of semantic categories.

In multi-frame network, given point clouds $\{\mathbf{P}_i, \ldots, \mathbf{P}_k\}$, We pass them through segmentation subnetwork and separate CVP of 3D branch and obtain augmented feature maps $\{\mathbf{A}_i', \ldots, \mathbf{A}_k'\}$ (see Figure 2(a)). Then we fuse them and get multi-frame fused augmented feature map $\mathbf{A}' \in \mathbb{R}^{L \times W \times H \times C'}$. We regard $\mathbf{A}'$ as augmented feature map $\mathbf{A}_i$ in a single-frame network, and the other modules of the multi-frame network are consistent with a single-frame network (see Figure 1(a)).

We set the branch number in the CVP of the single-frame network to be the same as the frame number in the multi-frame network. Moreover, we divide **Multi-Frame Knowledge Distillation** (MFKD) into two stages in Figure 2, we caculate the difference between $\{\mathbf{A}_i, \ldots, \mathbf{A}_k\}$ and $\{\mathbf{E}_i^0, \ldots, \mathbf{E}_i^{Q-1}\}$ correspondingly in stage-1 distillation to drive the branches in CVP of the single-frame network to learn the semantic feature distribution of corresponding frame and condense the various possibilities contained in the corresponding frame. We compute the difference between $\mathbf{A}'$ and $\mathbf{A}_i$ in stage-2 distillation to drive CVP in the single-frame network to learn multi-frame fused semantics further.

## 4 Architecture of VPNet

This section details the dual-branch VPNet with the confident voxel proposal (CVP) and the multi-frame knowledge distillation (MFKD).

### 4.1 Dual-branch Completion Network

As Figure 1(b) illustrates, VPNet has a 3D completion branch and a BEV completion branch, and we utilize multiple feature fusion schemes to combine them to achieve improved completion results.

**Feature Initialization** Given a point cloud $\mathbf{P}_i = \{(x_p, y_p, z_p, r_p) \mid p = 0, 1, \ldots, N_i - 1\}$. $x_p, y_p, z_p$ are the coordinates. $r_p$ is the reflectivity of point $p$, we update $\mathbf{P}_i$ as $\mathbf{P}_i'$ during voxelization. This update is represented as $\mathbf{P}_i' = \{(x_p, y_p, z_p, \delta x_p, \delta y_p, \delta z_p, r_p) \mid p = 0, 1, \ldots, N_i - 1\}$, where $\delta x_p$, $\delta y_p$, $\delta z_p$ are the differences in each dimension between the coordinates and the voxel center that point $p$ belongs to. Then, we initialize $\mathbf{V}_i$ and $\mathbf{B}_i$ as follows:

$$\{\mathbf{V}_i, \mathbf{B}_i\} = \{\mathcal{M}, \tilde{\mathcal{M}}\}(\mathcal{F}(\mathbf{P}_i')), \tag{1}$$

where $\mathcal{F}$ is a shared MLP that extracts the point-wise features, $\mathcal{M}$ and $\tilde{\mathcal{M}}$ indicate selecting the maximum from points that are in the same voxel or column.

**Dual-branch Completion** The 3D completion branch comprises a segmentation subnetwork and a completion subnetwork. The segmentation subnetwork, adapted from Cylinder3D [47], captures

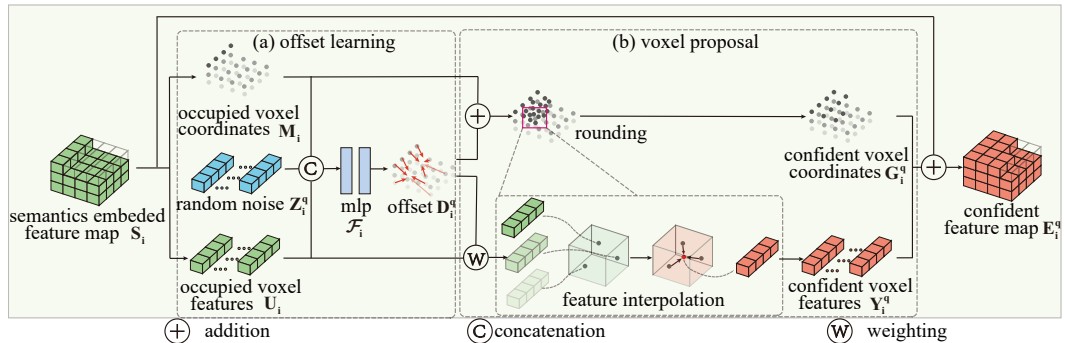

Figure 3: Branch $i$ of confident voxel proposal (CVP), we divide it into two steps: (a) offset learning and (b) voxel proposal.

semantic embedded feature map $\mathbf{S}_i$ from $\mathbf{V}_i$. These feature maps are fed into the completion subnetwork, which synthesizes the completed 3D feature map $\mathbf{F}_i^{3d}$. The completion subnetwork includes the proposed CVP module and several 3D dense convolution kernels of varying sizes.

In parallel, the BEV completion branch utilizes a 2D U-Net architecture. It reconstructs the scene from BEV. The BEV feature map $\mathbf{B}_i$ is processed through this branch to produce the completed BEV feature map $\mathbf{F}_i^{bev}$. With sum operation, we compress features extracted by the 3D segmentation subnetwork's encoder blocks along the height axis. We integrate the compressed feature maps to corresponding levels of BEV encoder blocks. This establishes an early fusion of 3D and BEV features that enhances the global perception capabilities of the 3D branch and the spatial analysis capabilities of the BEV branch. We utilize BEV-3D Fusion to generate the final completion result $\mathbf{R}_i$ as:

$$\mathbf{R}_i = \mathcal{C}(\mathcal{R}(\mathcal{I}(\mathbf{F}_i^{bev})), \mathbf{F}_i^{3d}), \tag{2}$$

where $\mathcal{I}$ is a convolution layer that increases the feature channels of $\mathbf{F}_i^{bev}$, $\mathcal{R}$ is a reshape operation that expands the height dimension from channels and $\mathcal{C}$ is concatenation along the channel dimension. This establishes the later fusion of 3D and BEV completion branches.

## 4.2 Confident Voxel Proposal

We propose confident voxels by offset learning and feature propagating from occupied voxels. We take $q^{th}$ branch as an example to describe the details of CVP.

**Offset Learning**   As illustrated in Figure 3, the segmentation subnetwork extracts a sparse semantics embedded feature map $\mathbf{S}_i$, from which we initialize the occupied voxel coordinates $\mathbf{M}_i = \{(x_j, y_j, z_j) \mid j = 0, 1, \ldots, J_i - 1\}$ and occupied voxel features as $\mathbf{U}_i \in \mathbb{R}^{J_i \times C'}$. In Figure 3(a), we initialize random noise $\mathbf{Z}_i^q \in \mathbb{R}^{J_i \times C_z}$, $C_z$ is the channel number of noise $\mathbf{Z}_i^q$ and $\mathbf{Z}_i^q \sim \mathcal{N}(0, 1)$. We computes a groups of offsets $\mathbf{D}_i^q \in \mathbb{R}^{J_i \times 3}$ for each coordinate in $\mathbf{M}_i$ as:

$$\mathbf{D}_i^q = \tilde{\mathcal{F}}_i(\mathcal{C}(\mathbf{M}_i, \mathbf{U}_i, \mathbf{Z}_i^q)). \tag{3}$$

$\tilde{\mathcal{F}}_i$ is the shared MLP in CVP. $\mathbf{M}_i$ allows the model to consider the geometric information of the partial scene, $\mathbf{U}_i$ introduces rich semantic context. The random noise $\mathbf{Z}_i^q$ drives the voxel coordinates $\mathbf{M}_i$ away from the initial position and ensures the robustness of offset learning. By sampling random noise across multiple branches of the CVP module, we generate various offset sets, enabling the inference of multiple semantic possibilities for the voxels. During offset learning, we employ occupied voxel coordinates $\mathbf{M}_g$ of completion ground truth as supervision.

**Voxel Proposal**   As shown in Figure 3(b), we propose coordinates that contain decimals by adding offsets $\mathbf{D}_i^q$ to initial coordinates $\mathbf{M}_i$ and compute confident voxel coordinates $\mathbf{G}_i^q$ with rounding operation on proposed coordinates. We formulate the feature propagation process as:

$$\mathbf{Y}_i^q = \mathcal{I}(\mathcal{W}(\mathbf{U}_i, \mathbf{D}_i^q)), \tag{4}$$

where $\mathcal{W}$ is the operation of updating feature $\mathbf{U}_i$ according to the offsets $\mathbf{D}_i^q$ that makes the features that propagate farther be pruned more. It ensures the reliability of feature propagation, $\mathcal{I}$ is feature

interpolation that computes the feature of the voxel center from proposed points in the same voxel, and $\bar{\mathbf{Y}}_i^q$ is the confident voxel features after propagation. We construct confident feature map $\mathbf{E}_i^q$ as:

$$\mathbf{E}_i^q = \mathbf{DDCM}(\mathcal{D}(\mathbf{G}_i^q, \mathbf{Y}_i^q) + \mathbf{S}_i), \tag{5}$$

where $\mathcal{D}$ is confident voxels with coordinates $\mathbf{G}_i^q$ and confident voxel features $\mathbf{Y}_i^q$. $\mathbf{DDCM}$ is a modified Dimension-Decomposition based Context Modeling [47] module that refines the features after propagation and reconstructs the semantic context.

As Figure 4 indicates, we adopt the weighted fusion strategy [48] and modify it to fuse the branches in CVP and compress the possibilities in branches. We compute the weights $\mathbf{W}_i^q \in \mathbb{R}^{1 \times C'}$ as:

$$\mathbf{W}_i^q = \mathcal{J}_i^q(\tilde{\mathcal{A}}(\sum_{q=0}^{Q-1} \mathbf{E}_i^q)). \tag{6}$$

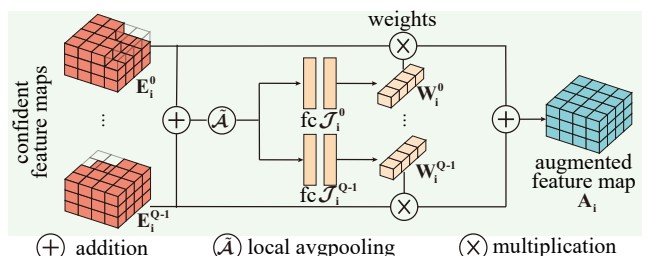

Figure 4: Architecture of the multi-branch fusion.

$\tilde{\mathcal{A}}$ is local average pooling that compresses the local representation for each branch. We flatten the pooled feature then, and $\mathcal{J}_i^q$ is the fully connected layer for $q^{th}$ branch. We fuse the branches in CVP according to the weights and get an augmented feature map $\mathbf{A}_i$.

### 4.3 Multi-Frame Knowledge Distillation

We construct a multi-frame network that proposes confident voxels and generates an augmented feature map for each frame with CVPs. We utilize MFKD to distill the semantic knowledge of augmented feature maps into a single-frame model in two stages to condense the voxel possibilities.

**Multi-Frame Network** As illustrated in Figure 1(a), we align the point cloud frames $\{\mathbf{P}_i, \dots, \mathbf{P}_k\}$ to the coordinate of the current frame $\mathbf{P}_i$. We input the aligned point clouds into the 3D completion branch separately. We get multi-frame augmented feature maps $\{\mathbf{A}_i, \dots, \mathbf{A}_k\}$. We fuse them as $\mathbf{A}'$ with the above-mentioned weighted fusion strategy. We regard $\mathbf{A}'$ as $\mathbf{A}$ in the single-frame network to build the multi-frame network. We input $\{\mathbf{V}_i, \dots, \mathbf{V}_k\}$ into 3D completion branch and only input $\mathbf{B}_i$ into BEV completion branch. We construct a multi-frame network that leverages multi-frame point clouds to model voxel semantic uncertainty with multiple CVPs from a 3D perspective.

**Multi-frame Distillation** We obtain augmented feature maps $\{\mathbf{A}_i, \dots, \mathbf{A}_k\}$ in multi-frame network and confident feature maps $\{\mathbf{E}_i^0, \dots, \mathbf{E}_i^{Q-1}\}$ in single-frame network, we calculate the difference $\mathcal{L}_{s1}$ between the corresponding feature maps with

$$\mathcal{L}_{s1} = \sum_0^{Q-1} \mathcal{E}(\hat{\mathcal{A}}(\mathbf{E}_i^q), \hat{\mathcal{A}}(\mathbf{A}_{i+q})) \tag{7}$$

where $\hat{\mathcal{A}}$ is a local average pooling function with kernel size $s \times s \times s$ to construct super voxels to meet the sparsity of features, and $\mathcal{E}$ is the Kull-

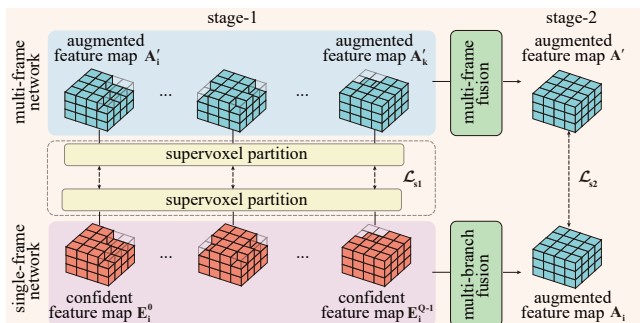

Figure 5: The overall architecture of MFKD. MFKD constructs two stages of distillation between CVPs of multi-frame networks and CVPs of the single-frame network.

back–Leibler divergence function. We build stage-1 distillation to drive the branches in CVP to simulate the knowledge learned by multi-frame CVPs and condense possibilities of the corresponding frame to branch in CVP.

**Training Losses** To improve the accuracy of semantic uncertainty modeling and reconstruct the scene details, given fused augmented feature map $\mathbf{A}'$ in the multi-frame network and augmented

feature map $\mathbf{A}_i$ in the single-frame network, we calculate the difference $\mathcal{L}_{s2}$ between them as:

$$\mathcal{L}_{s2} = \mathcal{E}(\mathbf{A}_i, \mathbf{A}'). \tag{8}$$

We avoid utilizing a super voxel partition here to prevent the blurring of features. Thus, we build stage-2 distillation to drive CVP in the single-frame network to simulate the knowledge after multi-frame CVPs. Finally, we achieve multi-frame distillation by joint stage-1 and -2 distillation as:

$$\mathcal{L}_{kd} = \mathcal{L}_{s1} + \mathcal{L}_{s2}. \tag{9}$$

We formulate total loss in the single-frame and multi-frame networks as:

$$\mathcal{L} = \alpha\mathcal{L}_{com} + \beta\mathcal{L}_{seg} + \gamma\mathcal{L}_{geo} + \delta\mathcal{L}_{kd}, \tag{10}$$

where $k = i$. $\mathcal{L}_{com}$ is completion loss. $\mathcal{L}_{seg}$ is segmentation loss. $\mathcal{L}_{geo}$ is geometry loss between proposed coordinates and coordinates ground truth. Here, we utilize Chamfer Distance [49]. $\alpha, \beta, \gamma$ and $\delta$ are weights of losses. We set $\alpha = 1.00$, $\beta = 0.10$, $\gamma = 0.01$ and $\delta = 0.50$ during distillation.

## 5 Experiments

### 5.1 Implementation Details

We implement VPNet with PyTorch[3] and train it on A6000 GPUs with a mini-batch of 8 for 80 epochs; we use Adam [50] optimizer with an initial learning rate of 0.001. We set feature map channel number $C' = 32$, random noise channel number $C_z = 4$, CVP branch number $Q = 3$, and super voxel partition kernel size $s = 4$.

### 5.2 Datasets and Metrics

We evaluate VPNet on SemanticKITTI [28] and SemanticPOSS [29] datasets, composed of real outdoor point cloud sequences. SemanticKITTI contains 22 sequences with 19 categories, 11/1/10 sequences for training/validation/online testing. SemanticPOSS contains six sequences divided into 11 categories; it contains 5/1 sequences for training/validation (testing).

According to SSCNet [1], we evaluate VPNet on Scene Completion (SC) with intersection-over-union (IoU), on Semantic Scene Completion (SSC) with IoU of each semantic category and mean of all semantic categories' IoU (mIoU).

### 5.3 Ablation Study of VPNet

In the ablation study, we conduct experiments on SemanticKITTI [28] validation set.

**Analysis of Network Framework**   We evaluate the impact of the BEV completion branch, segmentation subnetwork, completion subnetwork without CVP, and completion subnetwork with CVP. As Table 1 illustrated, in the first and second rows, we use the BEV branch and 3D branch without CVP separately and get (56.4% IoU, 22.2% mIoU) and (50.3% IoU, 23.6% mIoU).

In the third row, we use the BEV branch and segmentation subnetwork. Under the augmentation of 3D semantics, we get (58.3% IoU and 24.5% mIoU) which are much higher than the performances of the BEV branch and 3D branch separately. In the fourth row, we add a completion subnetwork without CVP to the network; this method produces (59.1% IoU, 24.9% mIoU) that improves the IoU with 0.8%. This proves the effectiveness of joint completion from different perspectives. The 3D branch riches the details (in the semantic

Table 1: Impact of dual-branch network components. "seg." means 3D segmentation subnetwork and "com." means 3D completion subnetwork.

| BEV | seg. | com. w/o CVP | com. w/ CVP | IoU | mIoU |
|:---:|:---:|:---:|:---:|:---:|:---:|
| ✓ | ✗ | ✗ | ✗ | 56.4 | 22.2 |
| ✗ | ✓ | ✓ | ✗ | 50.3 | 23.6 |
| ✓ | ✓ | ✗ | ✗ | 58.3 | 24.5 |
| ✓ | ✓ | ✓ | ✗ | 59.1 | 24.9 |
| ✓ | ✓ | ✗ | ✓ | **59.3** | **25.6** |

---

[3]https://pytorch.org/

aspect), while the BEV branch completes the scene coarsely and with higher completeness (from the geometric aspect).

We integrate CVP into the dual-branch network and achieve (59.3% IoU, 25.6% mIoU), a significant improvement of 0.7% mIoU compared to the network without CVP. This incremental improvement underscores the value of modeling the uncertainty of voxel semantics under the guidance of geometry for completion, marking a step forward in our understanding and application of these techniques.

**Internal Study of CVP**   We analyze the components of CVP in Table 2. In Table 2(a), we assemble CVP with random noise $\mathbf{Z}_i^q$ with different channel numbers. We get the best performance with $C_z = 4$. When less than 4, the learning of offsets is insufficient, and the semantic possibility learned from the voxel proposal is lacking. When more significant than 4, the random noise introduces too much meaningless information that adversely impacts the network's performance.

We propose confident voxels with multiple branches, so we analyze the impact of branch number $Q$ in Table 2(b). The network performance improves when the branch number is increased, and we get the best completion performance when we set the branch number to $Q = 3$. However, when we set $Q = 4$, we get similar performance (59.2% IoU, 25.6% mIoU) with $Q = 3$ as too many branches bring distractions to the network so we set $Q = 3$ during training.

The fusion strategy of multiple branches influences the performance of voxel semantic uncertainty modeling; we construct CVP with different fusion strategies and show the results in Table 2(c). Here, we set branch number $Q$ to 3. We compare the weighted fusion scheme with addition, concatenation, and average. These standard methods of feature fusion are weak in uncertainty modeling. Among the compared strategies, addition gets the best performance (59.3% IoU, 25.4% mIoU), but the weighted fusion we utilize still outperforms it with 0.2% mIoU. This demonstrates that weighted fusion models the voxel semantic uncertainty more accurately.

Table 2: Internal studies on random noise (a), branch number (b) and fusion strategy (c) of CVP.

(a) Channel number of noise.

| noise $C_z$ | IoU | mIoU |
|---|---|---|
| 0 | 59.0 | 25.0 |
| 2 | 59.1 | 25.4 |
| 4 | **59.3** | **25.6** |
| 6 | 58.8 | 25.3 |
| 8 | 58.5 | 24.9 |

(b) Branch number in CVP.

| branch $Q$ | IoU | mIoU |
|---|---|---|
| 0 | 59.1 | 24.9 |
| 1 | 59.0 | 25.1 |
| 2 | 59.2 | 25.4 |
| 3 | **59.3** | **25.6** |
| 4 | 59.2 | 25.6 |

(c) Fusion of branches in CVP.

| fusion strategy | IoU | mIoU |
|---|---|---|
| addition | 59.3 | 25.4 |
| concatenation | 59.2 | 25.2 |
| average | 59.0 | 25.3 |
| weighted fusion | **59.3** | **25.6** |

**Internal Study of MFKD**   As we build CVP with $Q = 3$, we implement a multi-frame network with three frames to build the distillation relationships between frame and branch in CVP correspondingly, and we present the results in Table 3. We get the best performance (61.1% IoU, 26.8% mIoU) with frames t/t+2/t+4, where t is the frame we use to train the single-frame. We get (60.2% IoU, 26.3% mIoU) with frames t/t+1/t+2

Table 3: Frames in multi-frame network.

| frames | IoU | mIoU |
|---|---|---|
| t / t / t | 59.5 | 25.6 |
| t / t+1 /t+2 | 60.2 | 26.3 |
| t / t+2 / t+4 | 61.1 | **26.8** |
| t / t+3 / t+6 | **61.4** | 26.6 |

as the adjacent frames contain insufficient supplementary information, and frames with larger intervals like t/t+3/t+6 have less guidance for modeling the uncertainty of semantics.

Table 4: Internal studies on stages of MFKD (a) and comparison with other distillation methods (b).

(a) Stages of MFKD. "voxel" means common voxel partition, "super-" means super voxel partition.

(b) Comparsion with other distillation methods.

| stage-1 (voxel) | stage-1 (super-) | stage-2 | IoU | mIoU |
|---|---|---|---|---|
| ✗ | ✗ | ✗ | 59.3 | 25.6 |
| ✓ | ✗ | ✗ | 59.5 | 25.5 |
| ✗ | ✓ | ✗ | 59.3 | 25.9 |
| ✗ | ✗ | ✓ | 59.6 | 25.8 |
| ✗ | ✓ | ✓ | **59.6** | **26.1** |

| | IoU | mIoU |
|---|---|---|
| KD [51] | 58.9 | 25.3 |
| PVKD [26] | 59.1 | 25.7 |
| DSKD [16] | 59.3 | 25.8 |
| MFKD (Ours) | **59.6** | **26.1** |

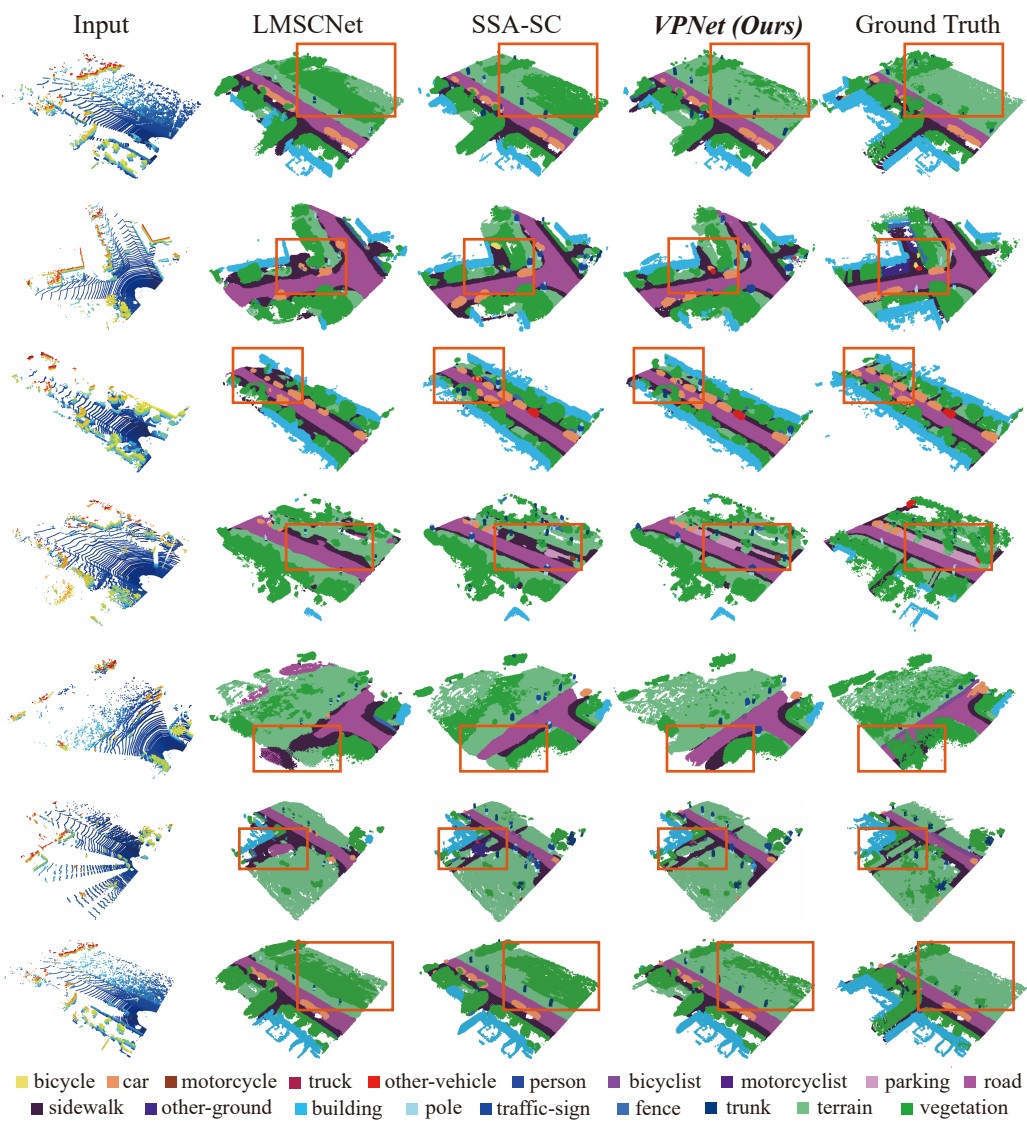

|  | Input | LMSCNet | SSA-SC | *VPNet (Ours)* | Ground Truth |

Figure 6: Completion results of different methods on SemanticKITTI validation set.

Legend: bicycle, car, motorcycle, truck, other-vehicle, person, bicyclist, motorcyclist, parking, road, sidewalk, other-ground, building, pole, traffic-sign, fence, trunk, terrain, vegetation

We conduct experiments using different distillation stages in Table 4(a). With stage 1 without super voxel partition, we get similar results (59.5% IoU, 25.5% mIoU) with the single-frame network, as ordinary distillation distracts the offset learning due to the neglect of sparsity. We add the super voxel partition to stage-1 distillation, and this method produces (59.3% IoU, 25.9% mIoU). We also build MFKD with stage-2 distillation only and get (59.6% IoU, 25.8% mIoU) that proves voxel-wise guidance like stage-2 is helpful to the semantic uncertainty modeling. We get better performance (59.6% IoU, 26.1% mIoU) with stage-1 with super voxel partition and stage-2 distillations that provide more accurate guidance. And we compare MFKD with other distillation methods in Table 4(b) where MFKD performs better than others and this proves the effectiveness of MFKD.

## 5.4 State-of-the-art Comparison

We compare our method with state-of-the-art methods on SemanticKITTI online testing set in Table 5. We present the visualization comparison in Figure 6. Our method outperforms other methods and demonstrates competitive performance. VPNet produces (60.4% IoU, 25.0% mIoU) that with 1.6% IoU and 1.5% mIoU improvement than SSA-SC [5] when training with single-frame without MFKD. It achieves (60.7% IoU, 25.6% mIoU) with 0.3% IoU and 0.6% mIoU improvement with MFKD.

Table 5: Comparison of VPNet with other works on SemanticKITTI online testing set.

| Method | IoU | mIoU | road | sidewalk | parking | other-ground | building | car | truck | bicycle | motorcycle | other-vehicles | vegetation | trunk | terrain | person | bicyclist | motorcyclist | fence | pole | traffic-sign |
|---|---|---|---|---|---|---|---|---|---|---|---|---|---|---|---|---|---|---|---|---|---|
| SSCNet [1] | 29.8 | 9.5 | 27.6 | 17.0 | 15.6 | 6.0 | 20.9 | 10.4 | 1.8 | 0.0 | 0.0 | 0.1 | 25.8 | 11.9 | 18.2 | 0.0 | 0.0 | 0.0 | 14.4 | 7.9 | 3.7 |
| SSCNet-full [1] | 50.0 | 16.1 | 51.2 | 30.8 | 27.1 | 6.4 | 34.5 | 24.3 | 1.2 | 0.5 | 0.8 | 4.3 | 35.3 | 18.2 | 29.0 | 0.3 | 0.3 | 0.0 | 19.9 | 13.1 | 6.7 |
| TS3D [52] | 29.8 | 9.5 | 28.0 | 17.0 | 15.7 | 4.9 | 23.2 | 10.7 | 2.4 | 0.0 | 0.0 | 0.2 | 24.7 | 12.5 | 18.3 | 0.0 | 0.1 | 0.0 | 13.2 | 7.0 | 3.5 |
| TS3D/DNet [28] | 25.0 | 10.2 | 27.5 | 18.5 | 18.9 | 6.6 | 22.1 | 8.0 | 2.2 | 0.1 | 0.0 | 4.0 | 19.5 | 12.9 | 20.2 | 2.3 | 0.6 | 0.0 | 15.8 | 7.6 | 7.0 |
| LMSCNet [2] | 55.3 | 17.0 | 64.0 | 33.1 | 24.9 | 3.2 | 38.7 | 29.5 | 2.5 | 0.0 | 0.0 | 0.1 | 40.5 | 19.0 | 30.8 | 0.0 | 0.0 | 0.0 | 20.5 | 15.7 | 0.5 |
| LMSCNet-SS [2] | 56.7 | 17.6 | 64.8 | 34.7 | 29.0 | 4.6 | 38.1 | 30.9 | 1.5 | 0.0 | 0.0 | 0.8 | 41.3 | 19.9 | 32.1 | 0.0 | 0.0 | 0.0 | 21.3 | 15.0 | 0.8 |
| Local-DIFs [53] | 57.7 | 22.7 | 67.9 | 42.9 | 40.1 | 11.4 | 40.4 | 34.8 | 4.4 | 3.6 | 2.4 | 4.8 | 42.2 | 26.5 | 39.1 | 2.5 | 1.1 | 0.0 | 29.0 | **21.3** | **17.5** |
| JS3C-Net [6] | 56.6 | 23.8 | 64.7 | 39.9 | 34.9 | 14.1 | 39.4 | 33.3 | 7.2 | 14.4 | 8.8 | **12.7** | 43.1 | 19.6 | 40.5 | 8.0 | **5.1** | 0.4 | 30.4 | 18.9 | 15.9 |
| SSA-SC [5] | 58.8 | 23.5 | 72.2 | 43.7 | 37.4 | 10.9 | 43.6 | 36.5 | **5.7** | 13.9 | 4.6 | 7.4 | 43.5 | 25.6 | 41.8 | 4.4 | 2.6 | 0.7 | 30.7 | 14.5 | 6.9 |
| Ours (w/o MFKD) | 60.4 | 25.0 | 72.4 | 44.3 | 40.5 | 14.8 | 44.0 | **37.2** | 4.3 | 14.0 | 9.8 | 8.2 | 45.3 | 30.9 | 42.1 | 4.9 | 2.0 | 2.4 | 32.7 | 17.1 | 8.8 |
| Ours (w/ MFKD) | **60.7** | **25.6** | **73.1** | **45.2** | **40.8** | **14.8** | **44.7** | 37.1 | 5.0 | **16.9** | **10.0** | 8.4 | **46.1** | **31.4** | **43.8** | **5.1** | 2.2 | **2.5** | **33.2** | 17.8 | 7.7 |

VPNet outperforms other methods in most of the semantic categories, and this demonstrates its effectiveness. In the first and fifth rows of Figure 6, VPNet achieves more complete geometry than other methods. It captures semantics more accurately (see the second and fourth rows). In the third and fifth rows, VPNet achieves complete and precise completion results for the distance of the scene thanks to the novel multi-frame distillation approach. We also validate the effectiveness of our method on the SemanticPOSS validation set and compare it with some representative methods in Table 6, and our method produces better completion performance (57.3% IoU, 23.3% mIoU) than other methods.

Table 6: Comparison of VPNet with other works on SemanticPOSS validation set.

| Method | IoU | mIoU | people | rider | car | traffic sign | trunk | plants | pole | fence | building | bike | road |
|---|---|---|---|---|---|---|---|---|---|---|---|---|---|
| SSCNet [1] | 41.7 | 12.1 | 8.2 | 0.4 | 1.3 | 1.0 | 3.7 | 34.4 | 4.8 | 3.3 | 34.3 | 16.0 | 25.6 |
| LMSCNet-SS [2] | 52.6 | 16.4 | 8.4 | 0.0 | 1.0 | 1.8 | 4.3 | 3.8 | 0.8 | 13.6 | 39.2 | 27.4 | 45.4 |
| SSA-SC [5] | 53.3 | 21.6 | **17.8** | 0.5 | **4.3** | 2.6 | 3.1 | 43.3 | 11.1 | 23.4 | 40.6 | 42.5 | 48.2 |
| Ours (w/o MFKD) | 56.9 | 22.4 | 15.1 | 0.4 | 1.7 | 1.0 | 4.9 | 46.4 | 9.4 | 28.0 | 43.2 | **45.0** | **51.0** |
| Ours (w/ MFKD) | **57.3** | **23.3** | 14.6 | **1.1** | 2.7 | **2.6** | **5.4** | **47.0** | **14.2** | **30.6** | **43.3** | 44.0 | 50.5 |

# 6 Conclusion

The recent progress in semantic scene completion has been achieved using the geometry and semantics of point clouds. Our paper introduces a dual-branch nework called VPNet with a confident voxel proposal that generates confident voxels through offset learning and multi-frame knowledge distillation that distills the possibilities from multi-frame to single-frame network. Our method has shown competitive performance on the SemanticKITTI and SemanticPOSS datasets.

# 7 Broader Impacts

VPNet enhances 3D perception capabilities by directly restoring geometric structures through voxel coordinate offset learning and conducting precise semantic feature propagation, thereby improving the ability for semantic scene completion. This work has inconspicuous negative societal impacts.

# 8 Limitations

The method encounters limitations in fine-grained geometric shape learning due to a single round of offset learning and feature propagation. An iterative process can solve this limitation.

# 9 Acknowledgement

This work was supported in part by the Key Science and Technology Program of the Ministry of Emergency Management of the People's Republic of China (2024EMST010102), in part by the Guangdong-Hong Kong Joint Funding for Technology and Innovation Grant (2023A0505010021), in part by the Hong Kong Polytechnic University Grant (P0048387), in part by the National Research Foundation, Singapore, and DSO National Laboratories under the AI Singapore Programme (AISG2-GC-2023-008), in part by Career Development Fund (CDF) of Agency for Science, Technology and Research (C233312028), in part by the National Research Foundation, Singapore, Infocomm Media Development Authority under its Trust Tech Funding Initiative (DTC-RGC-04).

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
