# OpenReview forum: "Voxel Proposal Network via Multi-Frame Knowledge Distillation for Semantic Scene Completion"
_NeurIPS.cc/2024/Conference — NeurIPS 2024 poster_

### Official Review · Reviewer_8T9a · 2024-06-30

**Soundness:** 3
**Presentation:** 2
**Contribution:** 3
**Rating:** 6
**Confidence:** 4

**Summary:**

This paper introduces Voxel Proposal Network which achieves semantic scene completion from both voxel and BEV perspectives. Beginning with confident voxel proposals, the information is propagated to other voxels, aiming to handle dynamic aspects. Multi-Frame Knowledge Distillation is also incorporated to accurately predict the voxel-wise labels.

**Strengths:**

1. Reconstructing from the reliable occupied voxels and propagating information from these voxels to others is a good solution for the semantic scene completion.
2. information distillation from multi-frames is also a normal strategy to improve the performance.
3. The ablation experiments in this paper are quite comprehensive.
4. The code provided in the appendix clearly demonstrates the details of the CVP module.

**Weaknesses:**

1. The motivation of the proposed components should be provided. At the beginning of the abstract, this paper mentioned that most previous methods use 3D/2D convolutions or attention mechanisms, having limitations in directly constructing geometry and accurately propagating features from related voxels. However, how can the proposed method tackle this problem？The authors are required to give a straightforward analysis.
2. The writing of this paper needs improvement. The author employs a single branch with multiple subnetworks for single-frame input, which seems processing the input sequentially. But line117 describes it using the “branch”, which can easily be misinterpreted as parallel. Besides, to my understanding, the multi-frame branch distill the knowledge of different timestamps to the results from different blocks, aiming to enhance the performance of feature propagation. It would better to give a overview before introducing the specific design of the network components.
3. The com. w/o CVP and com w/ CVP are confusing in the Table 1.
4. Comparison with other knowledge distillation algorithms (at least three algorithms) and the proposed MFKD is necessary, as done in the Table 4 in the SCPNet [1].
5. SCPNet also distills knowledge from multi-frame inputs to single-frame network, more analysis on the differences between MFKD and SCPNet is necessary.

Overall, the writing of this paper should be improved. Presenting motivation will help the readers understand the method better.

**Questions:**

Please see the weakness.

**Limitations:**

The authors adequately addressed the limitations and broader impacts. (line 464 - line 476)

---

> ### Author Rebuttal · Authors · 2024-08-02
>
> We sincerely thank you for your valuable comments on our paper's motivation, writing, and experiment. Below, we respond to your questions point-to-point.
>
> **Q1: At the beginning of the abstract, this paper mentioned that most previous methods use 3D/2D convolutions or attention mechanisms, having limitations in directly constructing geometry and accurately propagating features from related voxels. However, how can the proposed method tackle this problem?**
>
> **Answer:** The lost parts of the 3D shape lack sufficient information to infer the geometry and semantic labels. Thus, the completion model needs to regard the visible parts of the 3D shape as the cue to infer the lost parts. The existing models based on the deep networks typically use the convolution or attention scheme to learn the implicit correlation between the visible and lost parts from the training data. During the inference phase, the learned correlation helps the completion model to propagate the features of the visible parts to the underlying locations of the lost parts, whose geometry and semantic labels are inferred. Note that the lost details of the incomplete 3D shape are undetermined during the inference phase. **If the above feature propagation occurs between the visible parts and the locations without object occupancy, the inference of geometry and semantic labels likely fails**.
>
> We propose the Confident Voxel Proposal (CVP) to address the limitations of existing models. A CVP can be regarded as an explicit reconstruction of the lost details of the 3D shape. **What sets our approach apart from the existing methods is that we provide multiple CVPs for each frame, ensuring a more comprehensive shape reconstruction of the lost details**. These CVPs facilitate feature propagation between the visible parts and the confident locations with object occupancy, thereby significantly improving the completion result. Table 3 of the rebuttal PDF compares VPNet with and without the explicit learning of CVPs. This is done by using/removing the geometry loss $\mathcal{L}_{geo}$ in Eq.(10) of the paper. VPNet, with the explicit learning of CVPs, achieves IoU and mIoU better than the counterpart without the explicit learning of CVPs (see the first and second rows of Table 3 of the rebuttal PDF).
>
> **Q2: The author employs a single branch with multiple subnetworks for single-frame input, which seems processing the input sequentially. But line117 describes it using the "branch", which can easily be misinterpreted as parallel. Besides, to my understanding, the multi-frame branch distill the knowledge of different timestamps to the results from different blocks, aiming to enhance the performance of feature propagation. It would better to give an overview before introducing the specific design of the network components.**
>
> **Answer:** We will follow your suggestion to clarify the branches and add an overview of each network component, briefly describing its input and output.
>
> **Q3: The com. w/o CVP and com. w/ CVP are confusing in the Table 1.**
>
> **Answer:** By using "com. w/o CVP", we mean the 3D branch (represented by pink blocks) in Figure 1(a) works without the subsequent CVPs, while  "com. w/o CVP" means the 3D branch with CVPs. We will add this clarification to the manuscript.
>
> **Q4: Comparison with other knowledge distillation algorithms (at least three algorithms) and the proposed MFKD is necessary.**
>
> **Answer:** We follow your advice to compare the proposed MFKD with other knowledge distillation algorithms without changing the other parts of our framework. We compare MFKD with KD [Statistics 2015], PVKD [CVPR 2022], and DSKD [CVPR 2023] in Table 2 in the PDF along with the rebuttal. As shown in Table 2 of the rebuttal PDF, KD hurts the completion result because it only distills the features of the global scenes rather than proposals. PVKD and DSKD adopt distribution-based constraints but lack optimization before fusing voxel proposals across frames. In contrast, MFKD overcomes the above issues and achieves better completion results.
>
>
> **Q5: SCPNet also distills knowledge from multi-frame inputs to single-frame network. More analysis on the differences between MFKD and SCPNet is necessary.**
>
> **Answer:** Our MFKD contains core contributions different from those of SCPNet. The existing methods like SCPNet and MonoOcc with MFKD generally compute the visual feature of a single frame, where they fuse multiple frames’ features to capture the temporal information. They distill the knowledge of fused frames into the Student model. **During the above distillation process of the existing methods, the Student model misses the opportunity to learn to infer the lost details of each frame, which means the Student model only utilizes the available information of multiple frames**. We propose a novel MFKD scheme, which distills the knowledge of multiple frames and each frame’s inference of the lost details (named the Confident Voxel Proposal in our paper) into the Student model. The confident voxel proposals of different frames, which show consistent shapes, likely play as the overlapping objects across frames. They provide richer information for completing the 3D shape. Please also see our response to **Q1** of Reviewer Zma5 for more details on the motivation for using MFKD in semantic scene completion.
>
> Furthermore, a CVP explicitly reconstructs the lost details of the 3D shape. **Our approach provides multiple CVPs for each frame, ensuring a more comprehensive shape reconstruction of the lost details**. In contrast to other methods, CVPs facilitate feature propagation between the visible parts and the confident locations with object occupancy, thereby significantly improving the completion result. Please also see our response to **Q2** of Reviewer 2ocQ for the discussion on the importance of distilling the knowledge of CVPs for semantic scene completion.

---

> ### Comment · Reviewer_8T9a · 2024-08-08
>
> I appreciate the authors' response to my questions. Current manuscript lacks the valuable insights of the authors, which are more important for other researchers. It would better to update the overall manuscript in the revision. I'm willing to raise my score to weak accept later. Besides, I have the last question. As responded to the Reviewer Rcs7, VPNet achieves better performance than the SCPNet when using 3D convolutions. Will the authors release this part of code? (including training and evaluation)

---

> > ### Author Response · Authors · 2024-08-09
> > **Thanks for your suggestion!**
> >
> > Dear Reviewer 8T9a,
> >
> > We thank you again for your valuable comments, which will significantly help us to polish our paper. We will follow your suggestion to add the insights of proposing the multi-frame knowledge distillation and confident voxel proposal to the introduction section. We will also add overviews of these components to help readers to better understand our method.
> >
> > We are releasing the code package, including training and testing with 2D/3D convolution, via this anonymous GitHub repository: https://github.com/anonymous-github-VPNet/VPNet. Per your request, we are preparing a thorough README to explain how to install the necessary dependency and code package and configure the training/testing process.
> >
> > Best,
> >
> > Authors of Paper ID 4361

---

> > > ### Comment · Reviewer_8T9a · 2024-08-09
> > >
> > > Thanks, I've updated my score.

---

> ### Author Response · Authors · 2024-08-11
> **Thanks for Your Review**
>
> Dear Reviewer 8T9a,
>
> Thank you again for your review. We are pleased to see that the questions raised by you are solved.
>
> Best,
>
> Authors of Paper ID 4361

---

### Official Review · Reviewer_2ocQ · 2024-07-11

**Soundness:** 3
**Presentation:** 3
**Contribution:** 3
**Rating:** 5
**Confidence:** 4

**Summary:**

To directly construct scene geometry and accurately propagate features from relted voxels, the paper proposes VPNet with a voxel proposal mechanism to identify confident voxels for completion and a multi-frame knowledge distillation scheme to fuse information from multi-sweep LiDAR. VPNet achieves better performance than other methods for semantic scene completion on SemanticKITTI and SemanticPOSS.

**Strengths:**

1. For originality, VPNet predicts offsets to propose confident voxels which is distinct from existing methods.
2. For quality, the paper conducts extensive experiments on two benchmarks for semantic scene completion, together with ablation studies.

**Weaknesses:**

1. For clarity, the paper uses massive symbols to elaborate technical details which is hard to follow. I think there could be some abstraction.
2. For motivation, the paper claims that existing methods have problem with directly constructing scene geometry and accurately propagating voxel features, but I do not see why this is true and how VPNet is better in these aspects.

**Questions:**

Please give some explanations about the weaknesses.

**Limitations:**

The authors discuss the limitations of VPNet in the appendix but do not provide possible solutions.

---

> ### Author Rebuttal · Authors · 2024-08-02
>
> We sincerely thank you for your valuable comments on our paper's symbols and motivation. Below, we respond to your questions point-to-point.
>
> **Q1: For clarity, the paper uses massive symbols to elaborate technical details which is hard to follow. I think there could be some abstraction.**
>
> **Answer:** We recognize that complex symbols stem from multiple levels of superscripts and subscripts. We will trim the redundant superscripts and subscripts to simplify these symbols and equations.
>
>
> **Q2: For motivation, the paper claims that existing methods have problem with directly constructing scene geometry and accurately propagating voxel features, but I do not see why this is true and how VPNet is better in these aspects.**
>
> **Answer:** Please see our response to the second common question. The lost parts of the 3D shape lack sufficient information to infer the geometry and semantic labels. Thus, the completion model needs to regard the visible parts of the 3D shape as the cue to infer the lost parts. The existing models based on the deep networks typically use the convolution or attention scheme to learn the implicit correlation between the visible and lost parts from the training data. During the inference phase, the learned correlation helps the completion model to propagate the features of the visible parts to the underlying locations of the lost parts, whose geometry and semantic labels are inferred. Note that the lost details of the incomplete 3D shape are undetermined during the inference phase. **If the above feature propagation occurs between the visible parts and the locations without object occupancy, the inference of geometry and semantic labels likely fails**.
>
> We propose the Confident Voxel Proposal (CVP) to address the limitations of existing models. A CVP can be regarded as an explicit reconstruction of the lost details of the 3D shape. **What sets our approach apart from the existing methods is that we provide multiple CVPs for each frame, ensuring a more comprehensive shape reconstruction of the lost details**. These CVPs facilitate feature propagation between the visible parts and the confident locations with object occupancy, thereby significantly improving the completion result. Table 3 of the rebuttal PDF compares VPNet with and without the explicit learning of CVPs. This is done by using/removing the geometry loss $\mathcal{L}_{geo}$ in Eq.(10) of the paper. VPNet, with the explicit learning of CVPs, achieves IoU and mIoU better than the counterpart without the explicit learning of CVPs (see the first and second rows of Table 3 of the rebuttal PDF).
>
>
> **Q3: The authors discuss the limitations of VPNet in the appendix but do not provide possible solutions.**
>
> **Answer:** In the appendix, we introduce VPNet's limitations, which stem from inaccurate semantic segmentation and offset learning. These limitations lead to unreliable feature propagation, leaving much room for improving the completion result. We can equip VPNet with a more accurate segmentation subnetwork to alleviate the above limitations. Furthermore, we can iteratively conduct the offset learning and feature propagation, where the offset learning can be based on a more refined 3D shape. The extract computation during inference brought by the iteration can be saved by knowledge distillation.

---

> ### Author Response · Authors · 2024-08-11
> **Sincerely Request Your New Comment**
>
> Dear Reviewer 2ocQ,
>
> We thank you again for your valuable comments, which significantly help us to polish our paper. Could we kindly know if the responses have addressed your concerns and if further explanations or clarifications are needed? Your time and efforts in evaluating our work are appreciated greatly.
>
> Best,
>
> Authors of Paper ID  4361

---

> ### Author Response · Authors · 2024-08-13
> **Sincerely Request Your New Comment Again**
>
> Dear Reviewer 2ocQ,
>
> Again, please allow us to extend our sincere thanks to you, for your time and efforts of reviewing our paper. As the deadline for the authors' response is approaching, we sincerely request your comment on our primary response. This will definitely give us a valuable chance to address the questions unsolved.
>
> Best,
>
> Authors of Paper ID 4361

---

### Official Review · Reviewer_Rcs7 · 2024-07-12

**Soundness:** 2
**Presentation:** 3
**Contribution:** 2
**Rating:** 5
**Confidence:** 5

**Summary:**

The author introduced the Voxel Proposal Network (VPNet), a dual-branch semantic scene completion method with two key innovations.
First, the Confident Voxel Proposal (CVP) module, which includes offset learning and voxel proposal, generates a confident feature map based on the semantics-embedded feature map, enabling completion in the 3D branch. Second, the Multi-Frame Knowledge Distillation (MFKD) module distills semantic knowledge from each augmented feature map of the multi-frame network into the branches of the single-frame network in two stages, enhancing completion performance.

**Strengths:**

This manuscript has clear structure, well-benchmarked qualitative and quantitative results.

**Weaknesses:**

1. Lack of Comparison with State-of-the-Art Methods: The performance of VPNet is not compared with current state-of-the-art methods such as SCPNet. Specifically, (mIoU) achieved by VPNet is far lower than that of SCPNet. Or authors can argue and clarify clearly the protocol differences that render the comparison invalid.

2. SCPNet has already implemented multi-frame distillation to enhance performance. Thus, VPNet's use of Multi-Frame Knowledge Distillation (MFKD) is not readily qualified as a core contribution in the title. Or authors can dig into the effects of multi-frame distillation in different frameworks like SCPNet and [A].

[A] MonoOcc: Digging into Monocular Semantic Occupancy Prediction, ICRA 2024

**Questions:**

It is recommended to answer questions whether there is a protocol compatibility with SCPNet and whether there are unique insights about multi-frame distillation in this study over SCPNet and MonoOcc.

**Limitations:**

No social impact limitations to be addressed.

---

> ### Author Rebuttal · Authors · 2024-08-02
>
> We sincerely thank you for your valuable comments on our paper's experiments and core contribution. Below, we respond to your questions point-to-point.
>
> **Q1: Lack of Comparison with State-of-the-Art Methods: The performance of VPNet is not compared with current state-of-the-art methods such as SCPNet. Specifically, (mIoU) achieved by VPNet is far lower than that of SCPNet. Or authors can argue and clarify clearly the protocol differences that render the comparison invalid.**
>
> **Answer:** SCPNet utilizes many 3D convolution based subnetwork to complete the 3D shape, which makes the completion model complex while we only utilize simple convolution for refinement of CVP.  And as SCPNet indicates, this improve the mIoU greatly. Though SCPNet conducts the sparse convolution to save a portion of computation, the dense occupancy of voxel-wise features still makes the computation very expensive. Table 1 of the rebuttal PDF compares SCPNet (the first row) with our VPNet (the second row) concerning the model parameters (M), GPU memory (GB), inference speed (FPS), IoU, and mIoU. VPNet outperforms SCPNet regarding IoU, requiring fewer model parameters and much faster inference time. Because IoU indicates the accuracy of predicting the object occupancy, VPNet with a higher IoU shows a more robust power of inferring the geometry of the 3D shape than SCPNet.
>
> In contrast to IoU, we use mIoU to measure the accuracy of predicting the voxel-wise semantic labels of the 3D shape.  SCPNet yields a higher mIoU than VPNet, because 3D convolution better captures the semantic context of the objects. Though using 3D convolution with expensive computation contradicts our intention of using knowledge distillation to save model parameters and computation, we equip VPNet with more 3D convolutions after extracting point features and voxelization like SCPNet for a fair comparison. Table 1 of the rebuttal PDF shows that VPNet with 3D convolution (the third row) successfully outperforms SCPNet regarding mIoU, and it keeps similar model parameters with SCPNet.
>
>
> **Q2: SCPNet has already implemented multi-frame distillation to enhance performance. Thus, VPNet's use of Multi-Frame Knowledge Distillation (MFKD) is not readily qualified as a core contribution in the title. Or authors can dig into the effects of multi-frame distillation in different frameworks like SCPNet and MonoOcc.**
>
> **Answer:** Our MFKD contains core contributions that are different from those of SCPNet and MonoOcc. The existing methods like SCPNet and MonoOcc with MFKD generally compute the visual feature of a single frame, where they fuse multiple frames’ features to capture the temporal information. They distill the knowledge of fused frames into the Student model. **During the above distillation process of the existing methods, the Student model misses the opportunity to learn to infer the lost details of each frame, which means the Student model only utilizes the available information of multiple frames**. We propose a novel MFKD scheme, which distills the knowledge of multiple frames and each frame’s inference of the lost details (named the Confident Voxel Proposal in our paper) into the Student model. The confident voxel proposals of different frames, which show consistent shapes, likely play as the overlapping objects across frames. They provide richer information for completing the 3D shape. Please also see our response to **Q1** of Reviewer Zma5 for more details on the motivation for using MFKD in semantic scene completion.
>
> Furthermore, a CVP explicitly reconstructs the lost details of the 3D shape. **Our approach provides multiple CVPs for each frame, ensuring a more comprehensive shape reconstruction of the lost details**. In contrast to other methods, CVPs facilitate feature propagation between the visible parts and the confident locations with object occupancy, thereby significantly improving the completion result. Please also see our response to **Q2** of Reviewer 2ocQ for the discussion on the importance of distilling the knowledge of CVPs for semantic scene completion.

---

> ### Author Response · Authors · 2024-08-11
> **Sincerely Request Your New Comment**
>
> Dear Reviewer Rcs7,
>
> We thank you again for your valuable comments, which significantly help us to polish our paper. Could we kindly know if the responses have addressed your concerns and if further explanations or clarifications are needed? Your time and efforts in evaluating our work are appreciated greatly.
>
> Best,
>
> Authors of Paper ID 4361

---

> > ### Comment · Reviewer_Rcs7 · 2024-08-11
> > **Noted**
> >
> > I have read the rebuttal. The arguments are noted that in an enhanced setting the method can compare with SCPNet and the distillation differs from SCPNet and Monoocc in some ways. Now chasing 3DV deadline. Will update the score after a careful checking later: (1) the soundness of the new experiment; (2) the scientific value of the distillation setting. Since my two questions are both answered, I do not have further questions for now.

---

> ### Author Response · Authors · 2024-08-14
> **Sincerely Request Your New Comment Again**
>
> Dear Reviewer Rcs7,
>
> Again, please allow us to extend our sincere thanks to you, for your time and efforts of reviewing our paper. As the deadline for the authors' response is approaching, we sincerely request your further comment on our primary response. This will definitely give us a valuable chance to address the questions unsolved.
>
> Best,
>
> Authors of Paper ID 4361

---

> > ### Comment · Reviewer_Rcs7 · 2024-08-14
> > **BA**
> >
> > I have updated the recommendation to BA.
> >
> > I have two questions. They are answered.
> >
> > The authors should incorporate SOTA methods and clatification with SCPNet/MonoOCC into the manuscript. The current version that states MFKD as a new contribution without mentioning SCPNet/MonoOCC is not appropriate.

---

> ### Author Response · Authors · 2024-08-14
> **Thanks for your suggestion!**
>
> Dear Reviewer Rcs7,
>
> We thank you again for your valuable comments, which will significantly help us to polish our paper. We will follow your suggestion to add the comparison with SCPNet to the experiment section,  and add the insights of proposing MFKD and comparison with SCPNet/MonoOCC on distillation to the introduction section.
>
> Best,
>
> Authors of Paper ID 4361

---

### Official Review · Reviewer_Zma5 · 2024-07-16

**Soundness:** 3
**Presentation:** 2
**Contribution:** 3
**Rating:** 6
**Confidence:** 3

**Summary:**

This paper focuses on semantic scene completion. The authors propose a novel voxel proposal network and combine multi-frame knowledge distillation technique to reconstructs the scene geometry and implicitly models the uncertainty of voxel-wise semantic labels by presenting multiple possibilities for voxels. The proposed method has been proven effective on SemanticKITTI and SemanticPOSS datasets.

**Strengths:**

- The proposed method achieves state-of-the-art results on SemanticKITTI and SemanticPOSS datasets.
- The Voxel Proposal Network sounds interesting.

**Weaknesses:**

- My biggest concern is that the author's motivation for using MFKD is not clearly explained. Why use it to predict voxel-wise labels.
- Figure 3 is too complicated to understand.

**Questions:**

Can authors open source code？

**Limitations:**

Please refer to the weaknesses.

---

> ### Author Rebuttal · Authors · 2024-08-02
>
> We sincerely thank you for your valuable comments on the motivation and figure. Below, we respond to your questions point-to-point.
>
> **Q1: My biggest concern is that the author's motivation for using MFKD is not clearly explained. Why use it to predict voxel-wise labels?**
>
> **Answer:** Here, we introduce the motivation of MFKD for semantic scene completion. Suppose multiple frames of point clouds are available rather than a single frame only. In that case, people can leverage these sequential point clouds to complement each other and infer a complete 3D shape with semantic labels represented by voxels or point clouds. **Though this strategy is a natural technical choice, using multiple frames needs a complex model for learning the knowledge, including the visual pattern of a single frame and the temporal relationship between objects across various frames, substantially increasing the model complexity and inference time**. MFKD belongs to the family of knowledge distillation, where we transfer the knowledge learned by a complex Teacher model to a simpler Student model. Specifically for semantic scene completion, MFKD can distill the knowledge of multiple frames learned by the Teacher model into the Student model that only takes input as a single frame to infer the complete 3D shape quickly and accurately.
>
> The existing methods like SCPNet and MonoOcc with MFKD generally compute the visual feature of a single frame, where they fuse multiple frames’ features to capture the temporal information. They distill the knowledge of fused frames into the Student model. **During the above distillation process, the Student model misses the opportunity to learn to infer the lost details of each frame, which means the Student model only utilizes the available information of multiple frames**. We propose a novel MFKD scheme, which distills the knowledge of multiple frames and each frame’s inference of the lost details (named the Confident Voxel Proposal in our paper) into the Student model. The confident voxel proposals of different frames, which show consistent shapes, likely play as the overlapping objects across frames. They provide richer information for completing the 3D shape.
>
> Our paper follows the conventional protocol of semantic scene completion to represent the 3D shape as voxels for a fair comparison with other methods. MFKD can also help to predict point-wise labels in the 3D point cloud, which will be investigated in our future work.
>
>
> **Q2: Figure 3 is too complicated to understand.**
>
> **Answer:** We follow your comment to simplify Figure 3, the simplified figure is now included as Figure 1 of the rebuttal PDF.
>
> **Q3: Can authors open source code?**
>
> **Answer:** We release the code package via this anonymous Github repository **https://github.com/anonymous-github-VPNet/VPNet**.

---

> > ### Comment · Reviewer_Zma5 · 2024-08-11
> >
> > Thank the authors for their reply. The authors have addressed my concerns, so I have update my score.

---

> ### Author Response · Authors · 2024-08-11
> **Thanks for Your Review**
>
> Dear Reviewer Zma5,
>
> Thank you again for your review. We are pleased to see that the questions raised by you are solved.
>
> Best,
>
> Authors of Paper ID 4361

---

### Author Rebuttal · Authors · 2024-08-02

We sincerely thank all reviewers for their constructive comments. We provide a rebuttal PDF containing extra figures and tables to answer the common questions raised by reviewers. **The detailed explanations can be found in the point-to-point response to every reviewer**.

**Q1: Too complicated figure.**
- Reviwer Zma5: "Figure 3 is too complicated to understand."
- Reviewer 2ocQ: "For clarity, the paper uses massive symbols to elaborate technical details which is hard to follow. I think there could be some abstraction."

**Answer:** In Figure 1 of the rebuttal PDF, we provide a simplified Figure 3. We recognize that complex symbols stem from multiple levels of superscripts and subscripts. We will trim the redundant superscripts and subscripts to simplify these symbols and equations.


**Q2: Motivation of multi-frame knowledge distillation (MFKD) and comparison with other distillation methods.**
- Reviwer Zma5: "My biggest concern is that the author's motivation for using MFKD is not clearly explained."
- Reviewr Rcs7: "SCPNet has already implemented multi-frame distillation to enhance performance. Thus, VPNet's use of Multi-Frame Knowledge Distillation (MFKD) is not readily qualified as a core contribution in the title. Or authors can dig into the effects of multi-frame distillation in different frameworks like SCPNet and MonoOcc."
- Reviewer 8T9a: "Comparison with other knowledge distillation algorithms (at least three algorithms) and the proposed MFKD is necessary, as done in the Table 4 in the SCPNet. SCPNet also distills knowledge from multi-frame inputs to single-frame network, more analysis on the differences between MFKD and SCPNet is necessary."

**Answer:** In Table 1 of the rebuttal PDF, we compare VPNet with the state-of-the-art SCPNet in terms of the model parameters (M), GPU memory (GB), inference speed (FPS), IoU, and mIoU. In a fair setting, VPNet outperforms SCPNet. In Table 2 of the rebuttal PDF, we compare our MFKD with other knowledge distillation methods, KD [Statistics 2015], PVKD [CVPR 2022], and DSKD [CVPR 2023], where MFKD outperforms other methods.


**Q3: Confident Voxel Proposal (CVP) for geometry construction and feature propagation.**
- Reviewer 2ocQ: “For motivation, the paper claims that existing methods have problem with directly constructing scene geometry and accurately propagating voxel features, but I do not see why this is true and how VPNet is better in these aspects.”
- Reviewer 8T9a: “The motivation of the proposed components should be provided. At the beginning of the abstract, this paper mentioned that most previous methods use 3D/2D convolutions or attention mechanisms, having limitations in directly constructing geometry and accurately propagating features from related voxels. However, how can the proposed method tackle this problem? The authors are required to give a straightforward analysis.”

**Answer:** Table 3 of the rebuttal PDF compares VPNet with and without the explicit learning of CVPs. This is done by using/removing the geometry loss $\mathcal{L}_{geo}$ in Eq.(10) of the paper. VPNet, with the explicit learning of CVPs, achieves IoU and mIoU better than the counterpart without the explicit learning of CVPs (see the first and second rows of Table 3 of the rebuttal PDF).

---

### Decision · Program_Chairs · 2024-09-25

**Decision:**

Accept (poster)

**Comment:**

The paper presents valuable contributions to semantic scene completion. After the rebuttal, none of the reviewers have major concerns and the AC agrees that the paper can be accepted.